# Universal Successor Representations for Transfer Reinforcement Learning

**Chen Ma & Junfeng Wen**
Department of Computing Science, University of Alberta
chenchloem@gmail.com, junfengwen@gmail.com

**Yoshua Bengio**
MILA, Université de Montréal
yoshua.umontreal@gmail.com

## Abstract

The objective of transfer reinforcement learning is to generalize from a set of previous tasks to unseen new tasks. In this work, we focus on the transfer scenario where the dynamics among tasks are the same, but their goals differ. Although general value function (Sutton et al., 2011) has been shown to be useful for knowledge transfer, learning a universal value function can be challenging in practice. To attack this, we propose (1) to use universal successor representations (USR) to represent the transferable knowledge and (2) a USR approximator (USRA) that can be trained by interacting with the environment. Our experiments show that USR can be effectively applied to new tasks, and the agent initialized by the trained USRA can achieve the goal considerably faster than random initialization.

## 1 Introduction

Deep reinforcement learning (RL) has shown its capability to learn human-level knowledge in many domains, such as playing Atari games (Mnih et al., 2015) and control in robotics (Levine et al., 2016). However, these methods often spend a huge amount of time and resource only to train a deep model for very specific task. How to utilize knowledge learned from one task to other related tasks remains a challenge problem. Transfer reinforcement learning (Taylor & Stone, 2009), which reuses previous knowledge to facilitate new tasks, is appealing in solving this problem. Knowledge transfer would not be possible if the tasks are completely unrelated. Therefore, in this work, we focus on one particular transfer scenario, where dynamics among tasks remain the same and their goals are different, as will be elaborated in Sec. 2.

General value functions (Sutton et al., 2011) can be used as knowledge for transfer. However, learning a good universal value function approximator $V(s, g; \theta)$ (Schaul et al., 2015), which generalizes over the state $s$ and the goal $g$ with parameters $\theta$, is challenging. Unlike Schaul et al. (2015), who factorized the general state values into state and goal features to facilitate learning, we propose to learn a universal approximator for successor representations (SR) (Dayan, 1993), which is more suitable for transfer as we will see in Sec. 2.

Kulkarni et al. (2016) proposed a deep learning framework to approximate SR and incorporate it with Q-learning to learn SR by interacting with the environment on a single task. In comparison, our approach learns the universal SR (USR) that generalizes not only over the states but also over the goals, so as to accomplish multi-task learning and transfer among tasks. Additionally, we incorporate the framework with actor-critic (Mnih et al., 2016) to learn the SR in an on-policy fashion.

## 2 Universal Successor Representations

Consider a Markov decision process (MDP) with state space $\mathcal{S}$, action space $\mathcal{A}$ and transition probability $p(s'|s, a)$ of reaching $s' \in \mathcal{S}$ when action $a \in \mathcal{A}$ is taken in state $s \in \mathcal{S}$. For any goal $g \in \mathcal{G}$ (very often $\mathcal{G} \subseteq \mathcal{S}$), define pseudo-reward function $r_g(s, a, s')$ and pseudo-discount function $\gamma_g(s) \in [0, 1]$. $\gamma_g(s)$ can be that $\gamma_g(s) = 0$ when $s$ is a terminal state w.r.t. $g$. For any policy $\pi : \mathcal{S} \mapsto \mathcal{A}$, the general value function (Sutton et al., 2011) is defined as

$$V_g^\pi(s) = \mathbb{E}^\pi \left[ \sum_{t=0}^{\infty} r_g(S_t, A_t, S_{t+1}) \prod_{k=0}^{t} \gamma_g(S_k) \middle| S_0 = s \right]$$

For any $g$, there exists $V_g^*(s) = V_g^{\pi_g^*}(s)$ evaluated according to the optimal policy $\pi_g^*$ w.r.t. $g$. By seeing many optimal policies $\pi_g^*$ and optimal values $V_g^*$ for different goals, we would hope that the agent can utilize previous experience and quickly adapt to new goal. Ideally, such transfer would succeed if we can accurately model $\pi_g^*(s), V_g^*(s)$ using universal approximators $\pi(s, g; \theta_\pi), V(s, g; \theta_V)$ where $\theta_\pi, \theta_V$ are respective parameters. However, this would not be easy without utilizing the similarities within $r_g$ for all $g$, as we discuss next.

## 2.1 Transfer via Universal Successor Representations

We assume that the reward function can be factorized as (Kulkarni et al., 2016; Barreto et al., 2017)

$$r_g(s_t, a_t, s_{t+1}) = \phi(s_t, a_t, s_{t+1})^\top \mathbf{w}_g, \tag{1}$$

where $\phi \in \mathbb{R}^d$ are state features and $\mathbf{w}_g \in \mathbb{R}^d$ are goal-specific features of the reward. Note that if $\mathbf{w}_g$ can be effectively computed for any $g$, then we can quickly identify $r_g$ since $\phi$ is shared across goals. With this factorization, for a *fixed* policy $\pi$, the general value function can be computed as

$$V_g^\pi(s) = \mathbb{E}^\pi \left[ \sum_{t=0}^\infty \phi(S_t, A_t, S_{t+1}) \prod_{k=0}^t \gamma_g(S_k) \middle| S_0 = s \right]^\top \mathbf{w}_g = \boldsymbol{\psi}_g^\pi(s)^\top \mathbf{w}_g$$

where $\boldsymbol{\psi}_g^\pi(s)$ is defined as the *universal successor representations* (USR) of state $s$. The following Bellman equations enable us to learn USR the same way as learning the value function:

$$V_g^\pi(s) = \mathbb{E}^\pi[r_g(s, A, S') + \gamma(s)V_g^\pi(S')], \qquad \boldsymbol{\psi}_g^\pi(s) = \mathbb{E}^\pi[\phi(s, A, S') + \gamma_g(s)\boldsymbol{\psi}_g^\pi(S')].$$

**Framework Architecture.** In addition to modeling USR with a USR approximator (USRA) $\boldsymbol{\psi}^\pi(s, g; \theta_\psi)$ parametrized by $\theta_\psi$, we also model the policy with $\pi(s, g; \theta_\pi)$. Practically, we combine $\theta_\pi$ and $\theta_\psi$ in a deep neural network such that they share the first few layers and forked in higher layers. In order to quickly transfer to new goal, we need an efficient way to obtain $\mathbf{w}_g$ given goal $g$. This can be achieved by directly model $\mathbf{w}_g = \mathbf{w}(g; \theta_w)$ using a neural network. Finally, we further encode the state features $\phi(s, a, s')$ as $\phi(s, a, s'; \theta_\phi)$. More often, it is sufficient to model it as $\phi(s'; \theta_\phi)$ as we will do in the experiment. To summarize, the trainable parameters of our model are $(\theta_\pi, \theta_\psi, \theta_w, \theta_\phi)$, as shown in Fig. 1. **Transfer via USRA.** The trained USRA can be used (1) as an initialization for exploring new goal, and (2) to directly compute policy for any new goal.

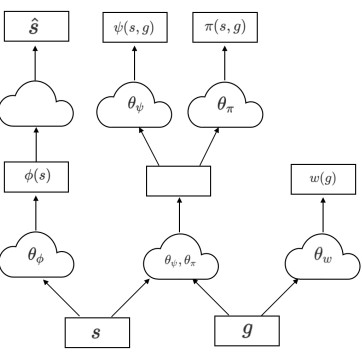

Figure 1: Model Architecture

## 2.2 Training USR

We begin with the state features $\phi(s)$. The state features are learned with an autoencoder, mapping from raw input $s$ to $\phi(s)$ and then back to $s$. In the early stage of the training, state $s$ are sampled from exploration of the agent with randomly initialized policy. The autoencoder are trained based on the reconstruction loss and $\theta_\phi$ are the encoder parameters. This step can be skipped in the case that $\phi(s)$ already has meaningful natural representations.

Once $\phi(s)$ is trained to converged, we then learn the rest of the parameters incorporated with actor-critic method by interacting with the environment. Algorithm 1 highlights the learning procedure. The update to $\theta_\pi$ is the typical policy gradient method (Williams, 1992).

## 3 Experiment

We perform experiments in a four-room grid-world environment. The agent's objective is to reach certain positions (goals). We use grid-world for simplicity, but our model uses raw pixels as input to show how USRA can handle continuous space. There are 64 goals in total, 48 of which act as source goals and the rest 16 as unseen target goals to be transfer to. An image indicating the agent's location is the input of the state. The goal is alike.

---

**Algorithm 1** USR with actor critic

---
1: **for** each time step $t$ **do**
2:     Obtain transition $\{g, s_t, a_t, s_{t+1}, r_t, \gamma_t\}$ from the environment following $\pi(s_t)$
3:     Perform gradient descent on $L_w = [r_t - \phi(s_{t+1})^\top \mathbf{w}(g; \theta_w)]^2$ w.r.t. $\theta_w$
4:     Perform gradient descent on $L_\psi = \|\phi(s_t) + \gamma_t \psi(s_{t+1}, g; \theta_\psi) - \psi(s_t, g; \theta_\psi)\|_2^2$ w.r.t. $\theta_\psi$
5:     Compute advantage $A_t = [\phi(s_t) + \gamma_t \psi(s_{t+1}, g) - \psi(s_t, g)]^\top \mathbf{w}(g)$
6:     Perform gradient descent on $J_\pi = \log \pi(s_t, g; \theta_\pi) A_t$ w.r.t. $\theta_\pi$,
7: **end for**

---

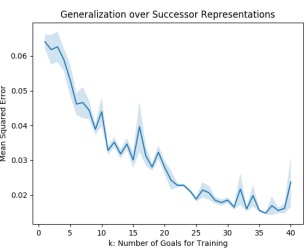 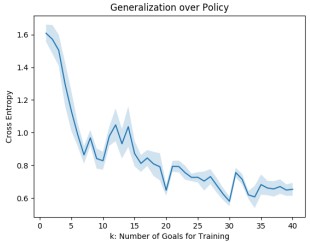 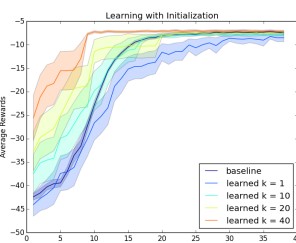

Figure 2: USR Generalization     Figure 3: $\pi$ Generalization     Figure 4: Effect of Initialization

### 3.1 GENERALIZATION PERFORMANCE ON UNSEEN GOALS

In this section, we show that how our model can generalize/transfer from source goals to target goals. Following our approach, we firstly trained USRA on $k$ source goals, randomly selected, until it converges. Then to measure the generalization performance on the target goals, we compute the distance between the USR/policy generated from our model to the "optimal" ones, which are obtained by learning directly on the target goals with the same model to convergence. Here we use Mean Squared Error (MSE) distance for USR, and cross entropy for policy with 6 repeats.

Fig. 2 and Fig. 3 visualize USR and policy's generalization performance w.r.t different numbers of source goals for training, with solid line as mean and shade as standard error. First note that as the number of source goals increases, the generalized policy and USR approach to the "optimal" ones. Second, the generalization performance trained on $k = 20$ goals is comparable to that on $k = 40$ goals. This indicates that only a relatively small portion of goals is required to achieve a decent generalization performance. These results demonstrate that our approach enables USR and policy to generalizes across goals.

### 3.2 TRAINED USRA AS INITIALIZATION

In this section, we show how the trained model can be used as an initialization for fast learning for target goals. We firstly train USRA on $k$ source goals, randomly selected, until convergence, then initialize the agent with this learned USRA for further exploration on target goals. Fig. 4 shows the average rewards the agent collected on target tasks over the steps. The baseline method is trained with random initialization. When the number of source goals $k$ is relatively small ($k = 1$), the agent learns more slowly than the baseline, which could be due to insufficient knowledge interfering with new goals' learning. However, when trained on a sufficient number of goals, 20/64 in this case, the agent can learn considerably faster for new goals. These results show that the agent initialized with trained USRA on only a small portion of the goals can learn much faster than random initialization.

## 4 CONCLUSION

In this work, we focus on solving transfer reinforcement learning problem in which the tasks share the same underlying dynamics but their goals differ. Our experiments show that the proposed USRA can generalize across tasks and can be used as a better initialization for learning new tasks.

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
