# OpenReview forum: "Universal Successor Representations for Transfer Reinforcement Learning"
_ICLR.cc/2018/Workshop — Accept_

### Official Review · AnonReviewer2 · 2018-02-25
**Good starting point**

**Rating:** 6
**Confidence:** 4

**Review:**

- A brief summary of the paper's contributions, in the context of prior work.
The paper studies how a policy can generalize between different goals in a maze, keeping the dynamics the same. Besides the standard actor-critic approach, the authors also learn a linear model for the reward, for a given goal. This approach is evaluated in a maze environment. Authors measure generalization by evaluating the distance between 1) trained policy + unseen goal 2) policy trained on that same goal.

- An assessment of novelty, clarity, significance, and quality.
The paper presents preliminary results, that welcome 1) more experiments + baseline comparisons in more challenging environments 2) analysis of what similarity between tasks the reward features are capturing and 3) more precise quantification of the minimal number of tasks to generalize well.

- A list of pros and cons (reasons to accept/reject).
pro: good direction
con:
- no analysis / visualization of task features
- no baselines (meta-learning approaches, hierarchical approaches)

---

### Official Review · AnonReviewer1 · 2018-03-07
**A nice idea for transfer learning in reinforcement learning domains with universal successor representations, however it fails to distinguish/compare existing work on transfer learning with success features.**

**Rating:** 5
**Confidence:** 4

**Review:**

Pros: 1) a very principled a framework for transfer reinforcement learning.
2) showing promising results
Cons:  1) there is a restriction here that the dynamics among tasks are the same and only goal location changes. It is unclear what is the motivation for such an assumption.
2)  lacking comparison to existing very related work.  The idea in this paper is very close to the following paper
Barreto et al. Successor Features for Transfer in Reinforcement Learning, NIPS 17
It might better if a comparison and more discussions regarding the differences between the proposed method and the above reference are provided.

---

### Official Review · AnonReviewer3 · 2018-03-10
**Accept**

**Rating:** 9
**Confidence:** 5

**Review:**

- The key idea is to factor the state-action or value function into expected discounted features and reward. This factorization has been explored in the past as the authors note but they generalize it for many goals. So in effect this is a very sound integration of two basic ideas -- successor representations and generalized value functions (GVFs or UVFAs). So USRs generalize SRs using GVFs/UVFAs. This is an important direction to explore and I encourage the authors to follow through with this work

- It would help to show visualizations of the goal space on the grid to get an intuition of the generalization

- It would also be interesting to see how few of source goals does it need to generalize on unseen goals

---

### Decision · Program_Chairs · 2018-03-20
**ICLR 2018 Workshop Acceptance Decision**

**Decision:**

Accept

**Comment:**

Congratulations, your paper was accepted to the ICLR workshop.